# *Citrus* Cell Suspension Culture Establishment, Maintenance, Efficient Transformation and Regeneration to Complete Transgenic Plant

**DOI:** 10.3390/plants10040664

**Published:** 2021-03-30

**Authors:** M. Moniruzzaman, Yun Zhong, Zhifeng Huang, Huaxue Yan, Lv Yuanda, Bo Jiang, Guangyan Zhong

**Affiliations:** 1Institute of Fruit Tree Research, Guangdong Academy of Agricultural Sciences, Guangzhou 510640, China; zhongyun99cn@163.com (Y.Z.); huangmo7721@163.com (Z.H.); jiangbo10086@126.com (B.J.); 2Key Laboratory of South Subtropical Fruit Biology and Genetic Resource Utilization, Ministry of Agriculture, Guangzhou 510640, China; yanhx628cn@163.com; 3Guangdong Provincial Key Laboratory of Tropical and Subtropical Fruit Tree Research, Guangzhou 510640, China; lvyuanda2015@163.com

**Keywords:** cell suspension, *Citrus*, tissue culture, de novo organogenesis, genetic engineering

## Abstract

*Agrobacterium*-mediated transformation of epicotyl segment has been used in *Citrus* transgenic studies. The approach suffers, however, from limitations such as occasionally seed unavailability, the low transformation efficiency of juvenile tissues and the high frequency of chimeric plants. Therefore, a suspension cell culture system was established and used to generate transgenic plants in this study to overcome the shortcomings. The embryonic calli were successfully developed from undeveloped ovules of the three cultivars used in this study, “Sweet orange”-Egyptian cultivar (*Citrus sinensis*), “Shatangju” (*Citrus reticulata*) and “W. Murcott” (*Citrus reticulata)*, on three different solid media. Effects of media, genotypes and ages of ovules on the induction of embryonic calli were also investigated. The result showed that the ovules’ age interferes with the callus production more significantly than media and genotypes. The 8 to 10 week-old ovules were found to be the best materials. A cell suspension culture system was established in an H+H liquid medium. Transgenic plants were obtained from *Agrobacterium*-mediated transformation of cell suspension as long as eight weeks subculture intervals. A high transformation rate (~35%) was achieved by using our systems, confirming BASTA selection and later on by PCR confirmation. The results demonstrated that transformation of cell suspension should be more useful for the generation of non-chimeric transgenic *Citrus* plants. It was also shown that our cell suspension culture procedure was efficient in maintaining the vigor and regeneration potential of the cells.

## 1. Introduction

The importance of *Citrus* spp. is linked to their enormous economic and nutritional values [1]. However, citrus cultivation has been confronting many challenges, including control of diseases [2,3,4,5]. Solutions to the problems will rely mostly on breakthroughs in breeding, which is also hindered by problems, such as sterility, self- and cross-incompatibility [6], widespread nucellar embryony, and long juvenile periods that are associated with traditional breeding practices [7].

Genetic engineering by transformation has been widely adopted for crop improvement [8,9,10], including citrus [11]. The main advantage of the technique is that it allows modification of interestingtrait(s) without altering the overall genetic makeup, which is useful in making desirable changes in elite cultivar(s) [12,13]. The common practice of citrus genetic transformation studies is *Agrobacterium tumefaciens* [14,15]-mediated transformation of epicotyl segments of in vitro-germinated seedlings [14,16,17]. However, seed availability is seasonal and genotype-dependent. For example, many citrus cultivars are seedless or few-seeded. In addition, genotypes showed a strong impact on citrus organogenesis and genetic transformation [18,19,20]. Notably, juvenile tissues from mandarins hybrids are more difficult to be transformed by *A. tumefaciens* [21,22], reducing seriously genetic transformation efficiency [15]. On the other hand, the use of mature materials for *A. tumefaciens*-mediated transformation could result in earlier fruit production, bypassing or reducing the juvenile phase [23,24,25]. However, mature tissues show recalcitrance for de novo organogenesis induction in tissue culture and have a high occurrence of chimeric transformation and losing transformed cell lines in transgenic plants [26].

Genetic transformation using embryogenic cell suspension cultures could be a better alternative for having higher organogenetic potential [27,28]. Regeneration of putatively transformed cells and subsequent grafting of transgenic micro-shoots on rootstocks may shorten the juvenile period for flowering and fruiting [29]. The classical conception of somatic embryogenesis (SE) is based on the unicellular origin of somatic embryos [30], and this mode of somatic embryo development was the most frequently noticed in embryogenic cell suspensions of *D. carota* [31]. However, both a multicellular and a unicellular origin of somatic embryos in the same regeneration system is quite a common phenomenon, as was observed in several species, including *Musa* spp. [32], *Cocos nucifera* [33], *Santalum album* and *S. spicatum* [34], and *H. vulgare* [35].

The “Sweet orange”-Egyptian cultivar, *Citrussinensis* (L.), is the most common and important species among *Citrus* [36].“Shatangju” (*Citrus reticulata*) [37,38] is a popular local mandarin and “W. Murcott”’ (*C. reticulata* Blanco x *C. sinensis* L. Osbeck) [39], a tangor of unknown parentage, is one of the main fresh cultivars in the world. By following the general procedure (establishment and maintenance of the cell suspension, transformation of the cells, and subsequent plant organogenesis from putative transgenic cells), we successfully established a cell suspension culture and an associated *Agrobacterium tumefaciens*-mediated transformation system for the three *Citrus* cultivars was also established. Finally, corresponding transgenic plants were recovered with high efficiency. The developed methods should be useful for *Citrus* genetic improvements through genome engineering experiments.

## 2. Results and Discussion

### 2.1. Embryogenic Callus Induction

The EME, DOG and H+H have commonly used media for somatic embryogenesis [28,40]. In this experiment, embryonic calli were successfully induced from ovules of all three cultivars (“Sweet orange”-Egyptian cultivar, “Shatangju” and “W. Murcott”) on all three different solid media (EME, DOG and H+H). As shown in Figure 1A, the highest callus induction occurred on EME medium, while the lowest was on H+H, although no statistically significant difference was found among the three media used in the study. However, in some other cases and other plant species, media have shown to have significant effects on callus induction [41].

Figure 1B is the callus induction rates of the 3 cultivars in the case of 8 to 10 weeks old ovules. The highest induction rate, around 74%, was from “Sweet orange”*, whereas the lowest, around 71%, was from “W. Murcott”. Previous studies used excised nucelli [42], abortive ovules [43], unfertilized ovules [44], undeveloped ovules [45,46], isolated nucellar embryos [47], juice vesicles [48], anthers [49], styles and stigmas [50], leaves, epicotyls, cotyledons and root segments of in vitro grown nucellar seedling [51] for somatic embryogenesis in *Citrus.* We chose undeveloped ovules as callus induction material because previous studies showed that undeveloped ovule is a preferable material for somatic embryogenesis not only for having higher regeneration capacity but also for being mostly virus-free [52]. Gmitter and Moore reported the explants regeneration percentage from undeveloped ovule was between 0% and 70%, depending on genotypes [45], but all the 3 genotypes used in the study showed a higher than 70% induction rate.

Embryonic callus induction is closely associated with the differentiation status of the material (ovule) used [40,53]. In our experiment, the age of ovules was indeed showed a significant influence on the embryonic callus induction. As shown in Figure 2, the callus induction percentage varied from around 41% to 74% across the whole age group used in the study. However, it was neither the younger nor, the older age groups, but the middle age group (8 to 10 weeks) was the best in terms of callus induction rate.

### 2.2. Suspension Cell Culture and Plant Regeneration

In this experiment, suspension cell culture for all three genotypes was established in liquid H+H medium, as previous studies demonstrated that the medium (H+H) was suitable for citrus cell suspension culture [28,54]. Maintenance of suspension culture involves regular subculture (every two to three weeks), which is laborious but important for subsequent experiments and plant regeneration [28,55]. In this experiment, we investigated factors affecting intervals of suspension cell subculture and subsequent embryo development. Our results showed that adding a smaller amount (1ml) of suspension cells to fresh media (~50 mL) could extend subculture intervals to 8 weeks without affecting the following embryo production rate (15~16 per plate) (Figure 3). However, there was a significant reduction in the number of embryos (~8) produced per plate when the same 1 mL of suspension cells was used from 2 weeks subculture intervals, perhaps from an insufficient founder population. When larger volumes of suspension cells were used in the subculture, the subculture intervals were proportionally shortened. For example, using 5 mL of inoculation volume reduced subculture intervals to 2 weeks since longer intervals significantly reduced embryogenic capacity, possibly a result of nutrient exhaustion. Three milliliters inocula in 50 mL fresh medium was good for regular experiments (Figure 4B). This allowed the cells to grow and ensured sufficient cells for the experiments. However, for the maintenance of suspension cells, 1 mL inocula in 50 mL fresh medium was suitable for its significantly extended subculture intervals.

In this study, *Agrobacterium*-mediated transformation and regeneration of suspension cellsderived from “Sweet orange”, “Shatangju” and “W. Murcott” were successfully accomplished (Figure 4). BASTA (20 mg/L) was added to the media to suppress the growth of nontransgenic cells. The transformation percentage was 32 to 35 and not significantly different among the cultivars (Table 1). Genetic transformation with desirable genes is an effective alternative for *Citrus* improvement [56,57,58,59]. Apparently, higher transformation efficiency is preferable since more transformants mean the chance of selecting an ideal transgenic line is high. In this regard, cell suspension culture is better than other materials, such as commonly used epicotyl segments prepared from in vitro germinated seedlings [15,21,22] and mature stem pieces [22] that normally showed a very low transformation rate (less than 10%).

### 2.3. Transgenic Plant Recovery and Molecular Analysis of Transgenic Plants

The BASTA-survived in vitro micro-shoots were propagated in two ways: grafted on rootstocks (Figure 4G) or rooted on RMAN medium and then planted on soil (Figure 4F,H). Both methods gave a survival rate of higher than 90%. However, the growth of the in vitro rooted shoots was poorer than the grafted (Figure 4H,G). All plants regenerated from selection pressure of 20 mg/L BASTA contained the transgenes, as shown by PCR analysis of leaf genomic DNA (Figure 5), indicating that a concentration of 20 mg/L BASTA was high enough to screen out the transformants in our case. Tissues from different organs, including the apical and the basal leaves and even roots from invitro rooted plants, were examined by PCR. It seemed that no chimeric plant was detected, demonstrating that the chances were very slim for single-cells and/or a very small group of cells to produce chimeras (Figure 4C,D). Additionally, no visual phenotypic changes were observed on all transgenic lines so far.

RT–PCR results showed that the *DMR6* in all three tested transgenic lines had a higher expression level than the control, and particularly, transgenic line 2 showed the highest expression level (10-fold) (Figure 6). This may be because of the insertion of different numbers of gene copies in different transgene lines. Transgene expression level depends on transgene copy number and/or site of gene integration [60,61,62,63]. Different copy numbers in different transgenic lines could lead to variable gene expression levels in independent transformants [64]. Gene silencing could be induced by transgene [65], but no silencing was observed in PCR-tested transgenic lines in this study.

## 3. Materials and Methods

### 3.1. Plant Materials and Embryonic Callus Formation

Young fruits (post-anthesis) of different age categories (4 to 6 weeks, 8 to 10 weeks and 12 to 14 weeks) of the “Sweet orange”-Egyptian cultivar (*Citrus sinensis*), “Shatangju” (*Citrus reticulata*) and “W. Murcott” (*Citrus reticulata*) were collected from orchard belongs to the Institute of Fruit Tree Research in Guangzhou, China. The following operations were done under a laminar flow hood. The fruits were surface sterilized by rinsing in 70% ethanol for 45 seconds, followed by immersion in 10% bleach for 15 min [41]. They were then cut open with a sterilized blade, and the ovules were collected. For embryonic callus induction, three different solid media, -EME [66], DOG [40], and H+H [28] were used (all media compositions are shown Appendix A). Sterilized media were poured into Petri dishes and allowed to cool and become solid; on the next day, 4 to 6 ovules were placed in. Seeded Petri dishes were transferred to the incubation room and incubated in dark conditions at 26 ± 2 °C until embryonic callus appeared. The growing ovules were subcultured to fresh media at 15 days intervals until appeared the undifferentiated embryonic callus.

### 3.2. Establishment of Cell Suspension Culture and Plant Organogenesis

To establish cell suspension culture, 1 to 2 g of undifferentiated callus was placed into a 125 mL Erlenmeyer flask containing 20 mL of H+H liquid media. The Erlenmeyer flasks were placed on a rotatory shaker at 135 rpm under a 16 h photoperiod (70 μmol m^−^^2^ s^−^^1^) at 26 ± 2 °C. After the first week, 10 mL of fresh H+H liquid media was added to each flask, and after one more week (second week), again extra 20 mL of fresh H+H liquid media was added to each flask. In total, four to five weeks were needed to establish the desired cell suspension culture. A feed-batch subculture was used to maintain the suspension cell culture. To investigate the effect of adding a variable amount of suspension cell into fresh media and different subculture intervals on organogenesis, we added 1 mL, 3 mL and 5 mL of cell suspension into 50 mL of fresh liquid media and practiced subculture at 2 w, 4 w, 6 w and 8 w. For plant regeneration, the cell suspension was initially plated on EME-malt solid media for embryo production. When small calli appeared after 4–6 weeks, they were transferred to EME 1500 media for germination and growth of the embryos. Four weeks later, the germinated embryos were transferred to B+ media for axis elongation. Healthy embryos were transferred to DBA3 media for shoot induction and growth. The in vitro culture condition was a 16 h photoperiod (70 μmol m^−^^2^ s^−^^1^) at 26 ± 2 °C. Finally, some micro-shoots were grafted on rootstocks, and some others were cultured on RMAN rooting media for root induction. For micro-shoot grafting, vigorously growing rootstocks, Ziyang xiangcheng (*Citrus junos* Sieb.ex Tanaka), special local citrus germplasm, were selected and transported into a glasshouse. In the late afternoon, healthy in vitro generated shoots (ensuring the shoots were not dehydrated before grafting) with at least four leaves were collected and bark-grafted on the rootstocks at the height of 30 cm above the ground, where the trunk diameter was around 1 cm thick. Transparent zipper plastic bags were inside sprayed with water and used to cover grafted individual micro-shoot to maintained high humidity. The plastic bag was removed when the growing shoot was hard enough to withstand the external environment (at least 3 weeks after grafting). In the case of acclimatization of rooted shoots on soil, the well-rooted shoots having at least four leaves were planted on peat moss and perlite (50:50) mixture wet media. The high humidity of the newly planted baby plants was also maintained as described above.

### 3.3. Vector Construction, Agrobacterium Transformation and Plant Regeneration

Total RNA was extracted from fresh leaves using an RNA fast extraction kit (product code: RP3202, BioTeke Corporation) according to the manufacturer’s instruction. The quality and quantity of the RNA weremeasured by NanoDrop™ 2000 (Thermo Scientific™) spectrophotometer. RNA quality was further assessed by agarose gel electrophoresis. cDNA was synthesized from total RNA by using a reverse transcription kit, the Evo M-MLV RT kit (code no: AG11711, Accurate Biotechnology (Hunan) Co. Ltd.), according to the manufacturer’s instruction. The gene *CsDMR6* was used for this experiment. The gene *DRM6* is associated with salicylic acid (SA) metabolism, mainly involved in the breakdown of SA, and a recent report revealed that DMR6–like genes were able to suppress immunity in *Arabidopsis* [67]. Mutation in the *dmr6* gene results in increased SA levels and enhances resistance in *Arabidopsis* and tomato [67,68,69]. A recent report showed that SA and methyl salicylate (MeSA) inhibited citrus canker caused by *Xanthomonas citri* [70]. *CsDMR6* overexpression construct was used to study the role of *CsDMR6* in response to citrus Huanglongbing (HLB) as one of the further goals of our experiments. The plasmid pFGC5941 (https://www.snapgene.com/resources/plasmid-files/?set=plant_vectors&plasmid=pFGC5941, accessed on 30 May 2021), is used as the backbone to construct *CsDMR6* overexpression vector, which has the following key features: a kanamycin resistance (kan^R^) gene for bacterial selection, a BASTA resistance (bar) gene for plant selection, a CaMV 35S promoter to drive the expression of the targeted gene. To construct the overexpression vector, the full-length *DMR6* was PCR-amplified from the cDNA using *DMR6* specific primers containing *Asc*I and *BamH*I restriction sites. The amplicon was purified by using a gel extraction kit (Product Code: D2111-01, Magen, Guangzhou, China) and inserted into the pGFC5941 vector at the chosen two restriction sites (*Asc*I and *BamH*I) using ClonExpress II one-step cloning kit (C112-01/02, Vazyme Biotech Co., Ltd.). The final vector construct was transformed into the *Escherichia coli* DH5α strain. The cloned *DMR6* gene was confirmed by sequencing. Finally, the overexpression vector was introduced into *Agrobacterium tumefaciens* strain EHA105 by the freeze–thaw method. PCR-positive clones were individually augmented by liquid culture, and one of them was used in the following transformation experiment. To further prepare the bacterial cells, 3 to 5 mL aliquot of overnight culture was added into 25 mL of fresh LB liquid media containing appropriate antibiotics and cultured for 3–4 h at 28 °C. The cultured cells were precipitated by centrifuge at 6000 rpm for 8 min at 25 °C. The bacterial cells were then resuspended in EME-sucrose liquid medium and diluted to an optical density (OD) of 0.3. Then, the bacterial solution was added to a 100 × 15 mm Petri dish containing 20 mL of citrus cell suspension and put under continuous and gentle agitation for 20 min (before adding bacteria, the liquid medium from cell suspension was drained off using a Pasteur pipette). The cells were then blotted on a sterile paper towel and transferred onto semisolid EME-sucrose medium supplemented with acetosyringone. Co-culture was performed in the dark at 25 °C for 5 days. The putative transgenic cells were cultured on regeneration media until plant regeneration was achieved (see the previous section). The relevant media were supplemented with 20 mg/L BASTA along with an appropriate antibiotic(s).

### 3.4. PCR and qPCR Analysis

Genomic DNA was extracted from fresh leaves of both putative transgenic plants as well as the nontransgenic (controls) plants and roots from in vitro rooted transgenic plants using aquick plant genomic DNA extraction kit (product code: N1192, Guangzhou Dongsheng Biotech Co., Ltd, Guangzhou, China). The extracted DNA was used as templates for PCR to confirm the presence of the transgenes. *CaMV35S* forward primer (CTATCCTTCGCAAGACCCTTC) and *MDR6* reverse primer (CCACTCAGGCACATACTTGT) as well as *Bar* (bacterial bialaphos resistance gene) specific forward primer (GATGAACAAAGCCCTGAA) and reverse primer (CCAAGATCAATAAAGCCAC) were used in PCR amplification. PCR mixture contained 1 µL template DNA (200 ng DNA/µL), 1 µL (10 µM) of each primer, 25 µL 2X Pro Taq Master Mix (code no: AG11109, Accurate Biotechnology (Hunan) Co. Ltd.), and 22 µL double-distilled water. PCR reactions were performed on the thermal cycler 2720 (Applied Biosystems, USA) programmed at an initial denaturation at 95 °C for 4 min, followed by 30 cycles at 94 °C for 30 s, 56 °C for 30 s, and 72 °C for 1 min and a final extension at 72 °C for 7 min. Amplified products were separated by electrophoresis in 1.5% (*w*/*v*) agarose gels (Biowest regular agarose G-10) in Tris-acetate-ethylenediaminetetraacetic acid, visualized by 10X DNA-loading buffer (code no: A0072, Accurate Biotechnology (Hunan) Co. Ltd.) staining, and photographed under UV (ultraviolet) light with a Gel Doc system (Bio-Rad Laboratories, Hercules, USA). Each sample was amplified at least twice to verify reproducibility.

For real-time PCR analysis, cDNA was synthesized (see the previous section), and the reactions were performed in a Step OnePlus real-time PCR system (Applied Biosystems) using iTaqTM Universal SYBR^®^ Green Supermix kit (Bio-Rad Laboratories, Hercules, USA) by following the instructions. The *DMR6*-specific qPCR primers (forward -CAGCAACCCATTTGTCGTCT, and reverse -AACTTTTACCCACCATGTCCAG) were designed using the Primer 3.0 online primer design program (http://primer3.ut.ee, accessed on 30 May 2021) and synthesized. Real-time PCR reaction was conducted 40 cycles at following conditions: hold- 95 °C for 30 s; cycle (PCR stage) 95 °C for 5 s, 60 °C for 20 s; cycle (melt carve stage) 95 °C for 15 s, 60 °C for 30 s; and dissociation- 95 °C for 15 s. Values were normalized against the *Actin* gene. qPCR data were analyzed by Applied Biosystems analytical software, and fold-change was calculated using the 2^−∆∆Ct^ method.

### 3.5. Data Recording and Analysis

For the data analysis (ANOVA) of culture media effect, all explants on the individual medium were considered as one group irrespective of genotypes; for example, explants of all three cultivars on the EME medium were considered as one group. The same thing happened in the case of the other two media. For the analysis (ANOVA) of genotype effect, all explants of the individual genotype on all three media were considered as one group; for example, all explants of “Sweet orange” cultivar on all three media were considered as one group. The same thing happened in the case of the other two cultivars. For the analysis (ANOVA) of ovule age effect, all explants of the individual age group from all three cultivars on all three media were considered as one group; for example, all explants of 4–6 week-old ovules from the three cultivars on the three different media were considered as the same group. Data on embryonic callus induction were recorded after 6–8 weeks of incubation. In the regeneration of suspension cells, the produced embryos were counted after 10 to 12 weeks of incubation on an EME-malt medium. To analyze the effect of inoculation volume and subculture interval, all embryos produced from all genotypes of an individual subculture interval (i.e., 2 W) of the same inoculation volume (i.e., 1 mL) were considered as one group.

Data were recorded from three replications for each of three cultivars. The statistical difference among the means was analyzed by Duncan’s multiplerange test using SPSS (version 23), and results were expressed as mean ± standard error of three independent experiments.

## 4. Conclusions

Cell suspension culture was successfully established from the embryonic callus of three citrus cultivars, “Sweet orange”, “Shatangju” and “W. Murcott”. The cell suspension materials were successfully used in the *Agrobacterium*-mediated transformation study, and a very high percentage of transgenic plants, confirmed by PCR, were generated. Detailed procedures and relevant data on the experiments were provided, which should find broader use in similar research.

## Figures and Tables

**Figure 1 plants-10-00664-f001:**
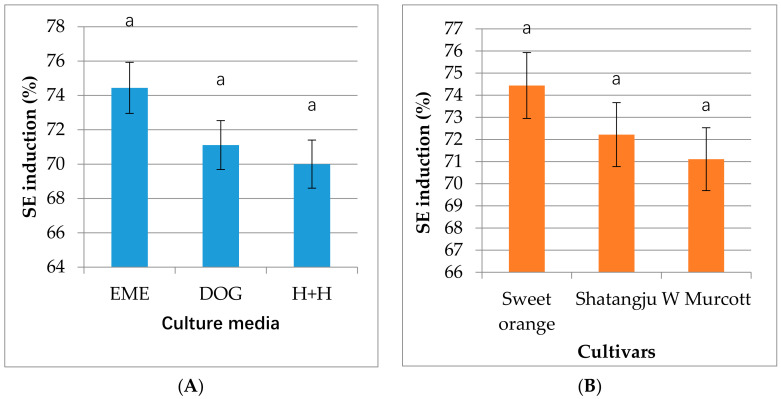
Somatic embryo induction.Data were recorded after 6–8 weeks of incubation. The statistical difference (*p* ≤ 0.05) among the means was analyzed by Duncan’s multiplerange test using Statistical Package for the Social Sciences (SPSS-version 23), and results were expressed as mean ± standard error of three independent experiments. (**A**) Culture media effect, all explants on the individual medium were considered as one group irrespective of genotypes (**B**) Genotypes effect, all explants of the individual genotype on all three media were considered as one group.

**Figure 2 plants-10-00664-f002:**
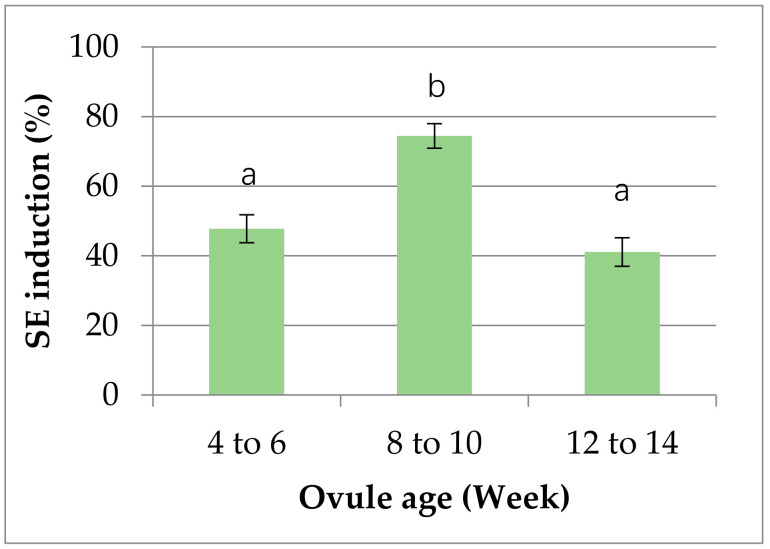
Effect of ovule age on embryonic callus induction. Data were recorded after 6–8 weeks of incubation. All explants of the individual age group (i.e., 4 to 6) from all three cultivars on all three media were considered as one group. The statistical difference (*p* ≤ 0.05) among the means was analyzed by Duncan’s multiplerange test using SPSS (version 23), and results were expressed as mean ± standard error of three independent experiments.

**Figure 3 plants-10-00664-f003:**
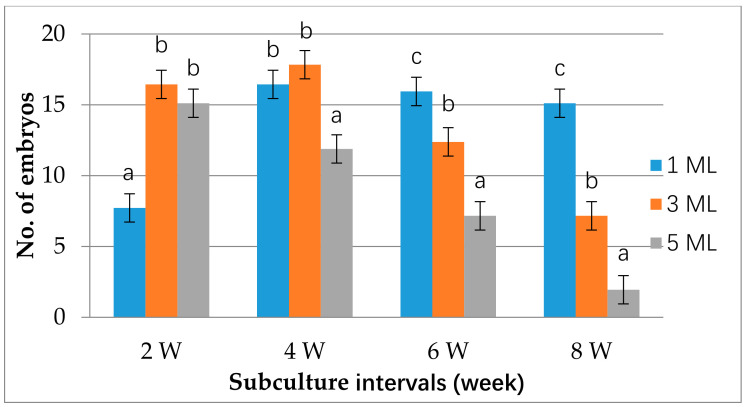
Effects of inoculation volume of suspension cells and subculture intervals on embryo production. The suspension cells were subcultured in H+H medium, and then the embryos were produced on EME-malt medium. The embryos were counted after 10 to 12 weeks of incubation on an EME-malt medium. All embryos produced from all genotypes of an individual subculture interval (i.e., 2 W) of the same inoculation volume (i.e., 1 mL) were considered as one group. The statistical difference (*p* ≤ 0.05) among the means was analyzed by Duncan’s multiplerange test using SPSS (version 23), and results were expressed as mean ± standard error of three independent experiments.

**Figure 4 plants-10-00664-f004:**
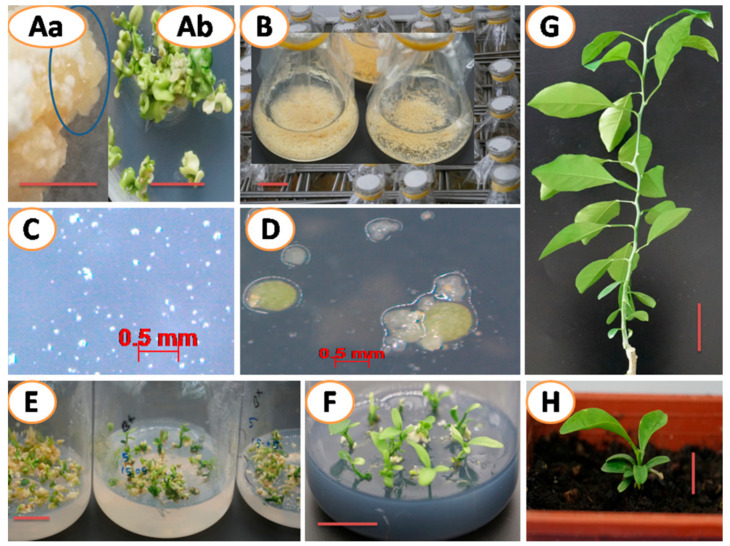
Embryonic callus induction, somatic embryo production, suspension cell culture establishment and plant regeneration of the “Sweet orange” (*Citrus sinensis*) cultivar. (**Aa**) Embryonic callus induction from 8 weeks old ovule on EME medium (**Ab**) and somatic embryo development, (**B**) Suspension cell culture establishment in an H+H medium, (**C**,**D**) Callus formation and embryo germination from suspension cell culture on an EME malt medium (the bars represent 0.5 mm), (**E**) Axis elongation of germinated embryos on an B+ medium, (**F**) Plants on RMAN rooting medium for root induction, (**G**) In vitro shoot grafted rootstock plant and (**H**) In vitro rooted plant transplanted on the soil. The bars represent 1 cm, except C and D.

**Figure 5 plants-10-00664-f005:**
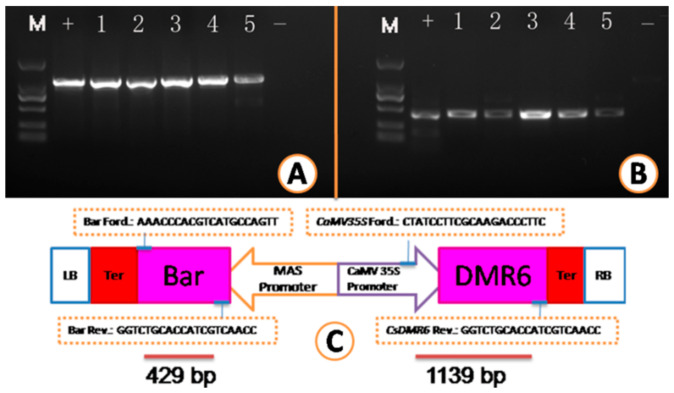
Gel electrophoresis of PCR amplified DNA from transgenic plants. 2000 kb DNA ladder (M), transgenic plants samples (1–5), positive control (+), negative control (−). (**A**) PCR amplicon (1139 bp) from CaMV35S forward and *CsDMR6* reverse primers.(**B**) PCR amplicon (429 bp) from *Bar* (bacterial bialaphos resistance gene) forward and reverse primers. (**C**) T-DNA constructs showing corresponding primer sites.

**Figure 6 plants-10-00664-f006:**
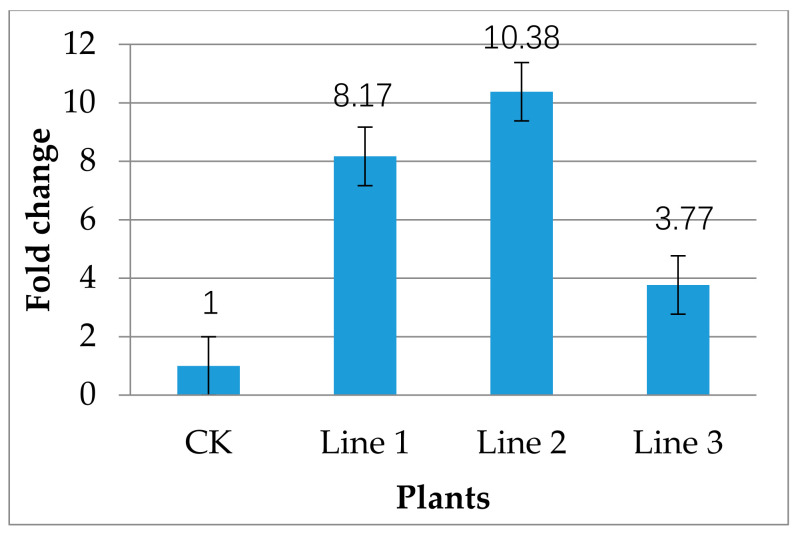
RT–PCR-mediated expression level analysis of *CsDMR6* in transgenic plants. Results were expressed as mean ± standard error of three independent experiments.

**Table 1 plants-10-00664-t001:** Transformation percentage of embryos developed from the cell suspension. Results were expressed as mean ± standard error of three independent experiments.

Cultivars	Embryos on Non-Selection Media	Embryos on BASTA (R) Selection Media	Transformation Percentage (%)
“Sweet orange”	51.94 ± 1.94	17.83 ± 0.39	35.09 ± 1.43
“Shatangju”	53.06 ±1.78	16.88 ± 0.63	32.42 ± 1.56
“W. Murcott”	53.66 ± 1.82	17.11 ± 0.8	32.38 ± 1.68

## Data Availability

All figures and tables are generated from these experiments. No materials form previous publication was used in our current manuscript.

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
