# Peer review of "Citrus Cell Suspension Culture Establishment, Maintenance, Efficient Transformation and Regeneration to Complete Transgenic Plant"

_plants, 2021, doi:10.3390/plants10040664_

Round 1

Reviewer 1 Report

The ms by Moniruzzaman et al describes the use of citrus suspension cells for Agrobacterium-mediated transformation. The authors managed to obtain transgenic shoots that were either grafted or rooted. The work seems interesting and may be useful for citrus breeders, but the description of the experimental approach and some of the results needs to be improved.

I have the following remarks:

On p3 l83 and l84 the authors calculate the callus induction rate and they present figures like 71.11% and 74.44%. This suggests an accuracy that is not met by the data as can be seen from the error bars in figure1. It would be more accurate to mention that rates were around 74% and around 71%.  And even this distinction is not statistically relevant. This applies similarly to other figures/tables such as table1.

In Figure4 the suspension is shown, but from the picture it cannot be seen how fine the suspension is in fact. Maybe for this procedure it does not matter so much, but some more details of the size of the cell clumps would be interesting for the reader. For instance size bars in figure 4c and D.

p4,5, table 1  and methods. The selection procedure is not described. Was Basta used as selective agent?

p5 Are the transgenic calli/embryos./shoots fully transgenic or chimeric?

p6 l152 what is gene DMR6 and why was this gene chosen for research?

p6 figure5 please show in a figure of the T-DNA where the primer sites are located.

p7 l206 What is vector pGFC5941. Please provide a reference or description.

In l 210 it is called pFGC5941, by the way.

Please describe here and/or in the text the vector, its gene content and selection marker.

p8 and figure5 what is HPT Hypoparathyroidism. A gene related to the thyroid gland? In plant papers HPT usually refers to hygromycin phosphotransferase.

Author Response

Thank you for your time and comments to improve our manuscript. We have tried to improve the manuscript according to the comment of you along with other reviewers. However we will be very happy to address any further comments, if required. Thank you again.

Reviewer 2 Report

the manuscript is of low interest since suspension cell culture for citrus is already well known and the genetic transformation protocol it has been already published since more than 10 years. 

The manuscript indicates three different cultivar but no cultivar is specified for  sweet orange. 

Author Response

Thank you for your time and comments to improve the readability of our manuscript. One of the objectives of this study were better management of cell suspension culture to provide year round supply of study material, improve the transformation efficiency and generalize the method for most cultivars. We believe this is very unique and it will contribute to our further experiments. We have tried to improve as per comments; however, we will be very happy for further improvement, if required.

Reviewer 3 Report

Dear authors, 
I have read your manuscript with great interest, and I congratulate you on your excellent work. 
However, the work needs some minor revisions before being accepted (you can find the revisions in the attached file). 
In addition, the discussion part needs to be thoroughly revised: it is poor in literature and yet the data needs to be better commented.

Best regards\

Author Response

Thank you for your time and comments to improve our manuscript. We have tried to improve the manuscript according to the comment from you along with other reviewers. However we will be very happy to address any further comments, if required. Thank you again.

Reviewer 4 Report

Manuscript "Citrus Cell Suspension Culture Establishment, maintenance, Efficient Transformation and Regeneration to Complete Transgenic Plant"by assumption it is an interesting research approach. However, it should be ensured that this valuable contribution is properly presented.

My comments about the manuscript in its current version are as follows:

  1. Authors, quite rightly, recall in lines 57-61 that SE may be either unicellular, or multicellular origin, however, unfortunately this was not checked in the course of the experiment. In fact, in the attached photographic documentation, we do not see any somatic embryos being produced. We were only shown (from a distance) the suspension culture, and this is definitely not enough.
  2. Plant material should be described appropriately. The reason for choosing the varieties is unknown. It was just that material was available, and not others? Or is the reason for choosing the material more important? The reader should be informed about this.
  3. The last sentence of the introduction is incomplete. It is generally not understood what the authors meant when writing this sentence. Please, specify the idea / intention.
  4. The issue of using three different media to initiate the culture of three cultivars is not reflected neither in Figure 1a (what genotype / material is presented?) nor in Figure 1b (what media was used?). In addition, the authors themselves state in lines 103-105 that the H + H medium was the most suitable, then why were the two others has been tested?
  5. Please think over the sentences on lines 107-116. What version would be proposed by the authors as optimal: 1 ml of suspension to 50 ml of fresh medium or 5 ml of inoculum / 50 ml of culture medium, or neither of them? These sentences do not explain this issue precisely.
  6. Please change the graph in Fig. 2 to a bar graph.
  7. Regarding the photo from the panel described as Figure 4. I would like to present the completion of the line allowing for the comparison of the order of magnitude between the objects in individual photos. Please show the embryos in photo A (along with documentation of the cross-section through embryogenic callus). Please also compare the height of the grafted rootstock with the plant transferred to the ground. The latter is much smaller, why?
  8. Since the authors obtained more than 90% of plants acclimatized (lines 140-141), please describe in detail how it was possible to achieve such a satisfactory result. The materials and methods only mention the substrate (194-195). This is too little information about the treatments used, allowing for acclimatization to ex vitro conditions.
  9. Please describe in the materials and methods the media used (also the one used for rooting)
  10. A huge number of errors can be found in References: lines "300, 310, 312, 314, 318, 321, 233, 326, 328, 329, 332, 334, 338, 344, 345, 250, 351, 355, 364, 367, 368, 270, 372, 374, 377, 380, 381-234, 386, 387, 389, 390, 393, 396, 398. Do the authors use only lowercase letters?
  11. Only when the text is corrected, please edit the abstract, as it will also require corrections.

I have tried to help you improve your manuscript as it is quite an important contribution to the improvement of Citrus materials. Please do your best during the improvement so that the readers know your contribution to the improvement of Citrus sinensis and Citrus reticulata breeding programmes.

Author Response

(The authors gave the same response as above.)

Round 2

Reviewer 2 Report

the manuscript contains some errors . The authors use the terms "cultivar" for sweet orange that is a specie and they do not indicate on which cultivar of sweet orange they did the experiment. in the text the authors indicate somatic embryogenesis as synonymous  of callus production.

Author Response

Response to Reviewer 2 (Second round)

Manuscript ID: plants-1129102-1

Title: Citrus cell suspension culture establishment, maintenance, efficient transformation, and regeneration to complete transgenic plant

Authors: M. Moniruzzaman * , Yun Zhong , Zhifeng Huang , Huanxue Yan , Lv Yuanda , Bo Jiang , Guangyan Zhong*

Thank you for your time and valuable comments to improve the readability of our manuscript. We have tried to improve as per comments; also tried to improve the English of the MS. However, we will be very happy for further improvement, if required.

Comment #1

the manuscript contains some errors . The authors use the terms "cultivar" for sweet orange that is a specie and they do not indicate on which cultivar of sweet orange they did the experiment. in the text the authors indicate somatic embryogenesis as synonymous  of callus production.

Response:

The sweet orange cultivar was ‘Egyptian sugar orange’ the Chinese name is as “埃及糖橙”. We provided the name in the revised version. We are sorry for misunderstanding of somatic embryogenesis with callus production. Throughout the manuscript we never talked nor mean callus induction/callus production as it generally means (unorganized mass of cells). As you know the general definition of Somatic embryogenesis is “Somatic embryogenesis is the process in which a single cell or a small group of cells follow a developmental pathway that leads to reproducible regeneration of non-zygotic embryos which are capable of producing a complete plant.” In this study, we have produced embryonic callus then fractionated the callus to produce single cell or a very small group of cells originated embryos. May be we called callus when single cell or a very small group of cells are growing for a chunk of cells to germinate the embryo. But if the sentences (text) are being read in context, we believe, it clearly means the embryogenesis. In revision we have tried to make it clearer. We have tried to improve the English as well. 

Reviewer 4 Report

Manuscript "Citrus Cell Suspension Culture Establishment, maintenance, Efficient Transformation and Regeneration to Complete Transgenic Plant" reports on research important to so called citriculture. The submitted version is certainly much better in comparison to the original text. Some errors still can be found and those need to be eliminated:

  1. Abstract,

line 1: ...transformationof".. the words should be separated: transformation of

The last sentence of Abstract: Please delete the word "more"

  1. Result and Discussion section

Figure 4 still need to be polished. What does the red line through the center of the photo panel mean? Some Figures are replicated. Why?

Figure 6: All charts should be unified in graphic design

References: There are still numerous errors. Write names in full or abbreviated form, it is not allowed to mix the forms.

Please correct all the errors in Your manuscript efficiently and accurately.

Author Response

Response to Reviewer 4 (Second round)

Manuscript ID: plants-1129102-1

Title: Citrus cell suspension culture establishment, maintenance, efficient transformation, and regeneration to complete transgenic plant

Authors: M. Moniruzzaman * , Yun Zhong , Zhifeng Huang , Huanxue Yan , Lv Yuanda , Bo Jiang , Guangyan Zhong*

Thank you for your time and valuable comments to improve the readability of our manuscript. We have tried to improve as per comments; we have tried to improve the English as well. However, we will be very happy to work on further improvement, if requires.

Comment: 1

Abstract: line 1: ...transformationof".. the words should be separated: transformation of

The last sentence of Abstract: Please delete the word "more"

Response: We are sorry these type of typos. We have corrected accordingly.

Comment: 2

Result and Discussion section

Figure 4 still need to be polished. What does the red line through the center of the photo panel mean? Some Figures are replicated. Why?

Response: We have tried to improve it. The PDF manuscript for reviews was showing new figure (improved) with old one as we used Microsoft office “Track Changes” to revise the original manuscript. This is also the same reason of showing replicated figures. The red line through the center of the photo panel means this plate is the deleted one.   

Comment: 3

Figure 6: All charts should be unified in graphic design

Response: We have tried to unify the graphic design for all Charts. May be, it would be more appropriate during production stage. 

Comment: 4

References: There are still numerous errors. Write names in full or abbreviated form, it is not allowed to mix the forms.

Response: We have gone through the references again, and improved as per instruction. We provided full name of the journals rather than abbreviation.